

# SDMtoolbox 2.0: the next generation Python-based GIS toolkit for landscape genetic, biogeographic and species distribution model analyses

Jason L. Brown,  Joseph R. Bennett and  Connor M. French

Department of Zoology, Cooperative Wildlife Research Laboratory, Southern Illinois University at Carbondale, Carbondale, IL, USA

## ABSTRACT

SDMtoolbox 2.0 is a software package for spatial studies of ecology, evolution, and genetics. The release of SDMtoolbox 2.0 allows researchers to use the most current ArcGIS software and MaxEnt software, and reduces the amount of time that would be spent developing common solutions. The central aim of this software is to automate complicated and repetitive spatial analyses in an intuitive graphical user interface. One core tenant facilitates careful parameterization of species distribution models (SDMs) to maximize each model's discriminatory ability and minimize overfitting. This includes carefully processing of occurrence data, environmental data, and model parameterization. This program directly interfaces with MaxEnt, one of the most powerful and widely used species distribution modeling software programs, although SDMtoolbox 2.0 is not limited to species distribution modeling or restricted to modeling in MaxEnt. Many of the SDM pre- and post-processing tools have 'universal' analogs for use with any modeling software. The current version contains a total of 79 scripts that harness the power of ArcGIS for macroecology, landscape genetics, and evolutionary studies. For example, these tools allow for biodiversity quantification (such as species richness or corrected weighted endemism), generation of least-cost paths and corridors among shared haplotypes, assessment of the significance of spatial randomizations, and enforcement of dispersal limitations of SDMs projected into future climates—to only name a few functions contained in SDMtoolbox 2.0. Lastly, dozens of generalized tools exists for batch processing and conversion of GIS data types or formats, which are broadly useful to any ArcMap user.

## INTRODUCTION

SDMtoolbox is a Python-based ArcGIS toolbox for spatial studies of ecology, evolution and genetics. SDMtoolbox consists of a series of Python scripts (79 and growing) designed to automate complicated spatial analysis in ArcMap (*ESRI, 2017*) and Python. Since SDMtoolbox's first release, in April 2014 (*Brown, 2014*), the program has been download over 30,000 times by users in 160 countries (from every sub-continent) and cited over

Corresponding author
Jason L. Brown, jason.brown@siu.edu, sdmtoolbox.help@gmail.com

**Table 1  Major differences between SDMtoolbox V1 and V2.**

| Feature | SDMtoolbox V1 | SDMtoolbox V2 |
|---|---|---|
| Compatibility with ArcGIS 10.3-10.5 | | X |
| Input Parameters Output As File | | X |
| Improved user performance, Python code is optimized, expanded user-guide | | X |
| Complete compatibility with the new open source version of Maxent (version 3.4) | | X |
| Total Tools | 59 | 79 |

180 times. Surveying SDMtoolbox's citations, most users (87%) used this toolkit for the preparation of SDM files, running SDMs and processing of SDM results. Since the first publication (*Brown, 2014*), SDMtoolbox has been under continuous development and expansion. SDMtoolbox remains a free, comprehensive Python-based toolbox for macroecology, landscape genetic and evolutionary studies to be used with ArcGIS 10.0 (or higher) with a Standard or Advance License and the Spatial Analyst extension. The toolkit simplifies many GIS analyses required for species distribution modelling and other spatial ecological analyses, alleviating the need for repetitive and time-consuming climate data pre-processing and post-SDM analyses.

## METHODS

SDMtoolbox is written in Python (v2.7) and incorporates Python libraries from ArcPy (ArcGIS 10.0-10.5), NumPy and SciPy that are incorporated into a single toolbox for use by ArcGIS 10.0-10.5 users. One tool, the *Run MaxEnt: Spatially Jackknife* tool, outputs java code batch file format to run the MaxEnt program.

## RESULTS

After the release of SDMtoolbox v1, we updated 28 (of 59) of the original tools to improve user performance and maximize compatibility with newer versions of ArcGIS 10.3-10.5 and the recent open source version of MaxEnt (version 3.4 or higher; *Phillips, Dudík & Schapire, 2017*). For several tools this required completely recoding the analysis pipeline (Table 1). In addition to updating many tools, we provide 20 new tools, increasing the total tools to 79 (Table 2). The adaption of the code in the *Run MaxEnt: Spatially Jackknife* tool for MaxEnt 3.4 enables output models in cloglog format and enabling/disabling use of threshold feature class in spatial jackknifing process. What follows are brief overviews of the new tools, organized by major tool category.

### Biodiversity measurements

This suite of tools calculates spatial biodiversity patterns. One of two new tools facilitates a quantitative method for locating hotspots of endemism, and is called Categorical Analysis of Neo- and Paleo-Endemism (CANAPE; *Mishler et al., 2014*). These analyses are able to classify neo-endemic and paleo-endemic species, young taxa and old taxa with restricted distributions, respectively. This method assesses the significance of branch lengths among

**Table 2  New Tools in SDMtoolbox v2.0.**

| Tool subgroup and name | Function | Numbers of tools |
|---|---|---|
| **Biodiversity Measurements** | | |
| • CANAPE categorization | • Runs categorizations of neo- and paleo-endemism on grids output from Biodiverse | 1 |
| • Quickly reclassify significance from randomizations | • Uses data from Biodiverse to randomize and reclassify significance | 2 |
| **Landscape Connectivity** | | |
| • Create Pairwise Distance Matrix | • Creates distance matrices showing both the least-cost-path (LCP) and the along-path-cost of the LCP | 1 |
| **SDM Tools** | | |
| • Split binary SDM by input clade relationship | • Splits a binary SDM by input user clade relationships. Can only be done with 2–10 clade Groups | 1 |
| • Sample by Buffered Local Adaptive Convex-Hull | • Limits selection of background points to area inside a buffered regional convex-hull created through species occurrences | 1 |
| **Basic Table, Shapefile, and Raster Tools** | | |
| • Project Shapefiles to User Specified Projection (folder) | • Projects entire folder of shapefiles to any input projection | 1 |
| • Define Projection (folder) | • Used to define the projection of any input (shapefile or raster) | 5 |
| • Polygon to Raster (folder) | • Converts polygon input into a raster format | 1 |
| • NetCDF to Raster (folder) | • Converts all NetCDF (.nc) files to raster | 1 |
| • Define NoData Value (folder) | • Redefines NoData value in rasters. Used to fix an error when creating rasters where the NoData value is changed | 1 |
| • Advance Upscale Grids (folder) | • Upscales all grids in folder to a coarser resolution | 1 |
| • Export JPEGs of all open files | • Exports JPEGS of all files in the map viewer | 1 |
| • Export Images of All Color Permutation of a RGB raster | • Exports images of all color permutations of a RGB raster | 1 |
| • Sample raster values at input localities (folder) | • Samples the values of TIFF rasters at the locations input. Allows field names to be up to 50 characters | 1 |
| • Increase Raster Extent/Snap All Raster to Same Extent (folder) | • This tool will increase or decrease spatial extent of all input rasters | 1 |

taxa that are either significantly shorter (neo) or significantly longer (paleo) than other areas in the landscape. The randomizations are performed in the standalone program Biodiverse (*Laffan, Lubarsky & Rosauer, 2010*) and outputs are input into SDMtoolbox 2.0, which performs the series of analyses for categorization of significant neo- and paleo-endemic areas. A second, similar tool categorizes significance of randomizations done in Biodiverse and is applicable to any randomization performed in Biodiverse (*Laffan, Lubarsky & Rosauer, 2010*).

## Landscape connectivity

This suite of tools measure landscape connectivity among populations. One prevalent method to do this is to estimate least-cost paths (LCPs) among sites (e.g., *Ray, 2005*; *McRae & Beier, 2007*). The new tool added to this category measures LCPs and outputs distance matrices of the LCP distance and the along-path-cost of each LCP.

## Species distribution modeling tools

Many of the Python scripts contained in SDMtoolbox were initially written for species distribution modelling. Two new tools have been added to this group. The first tool splits a binary SDM by user- input clade relationships. This is done by dividing the landscape by Voronoi polygons generated from input localities and then each polygon is grouped by the clade relationship. The polygon of each clade relationship is used to mask the input SDM and output is the proportion of the SDM corresponding to that clade distribution. This tool is useful for dividing species distributions by their phylogenetic relationships.

The second species distribution modeling tool provides a novel method for creating bias files for use in MaxEnt (see discussion for overview and importance of bias files in MaxEnt). Often distributions can be largely separated by unsuitable habitat. The second tool limits the selection of background points to an area encompassed by a buffered regional convex-hull based on species occurrences. The area of background selection is intermediate between the *buffered minimum-convex polygon* tool (which can include considerable area between distance localities) and the *distance from observed localities* tool (which can be quite restrictive). A main parameter for this tool is the alpha parameter that depicts the distance where points are aggregated into a convex-hull. Using this tool, bias files can represent several disjunct, buffered polygons. Larger values will result in areas of background selection more similar to a buffered minimum-convex polygon (MCP) and smaller values more similar to outputs from the *distance from observed localities* tool. Generally speaking, this tool typically results in regional buffered MCP based on spatial clusters of points.

## Basic tools

Most of the basic tools facilitate batch processing or conversion of data required for spatial analyses. Fourteen new tools have been added to this group. Given a long standing ArcMap issue associated with extracting raster values to points in ArcGIS, we created a tool that achieves the same end product using a different data pipeline that works in most cases where the ArcMap's native tool failed. Several of the new tools facilitate conversion of rasters to other raster formats (i.e., NetCDF or float values). Other new tools batch project or define the projection of input rasters and shapefiles to any projection. Another set of tools facilitate batch exportation of images or aids in the display of red-green-blue (RGB) bands in multiband rasters (e.g., which band is applied to red spectrum) by exporting images of all color permutations of RGB band combinations.

The default method for upscaling rasters (making a raster a coarser spatial resolution) is to resample by selecting the value of a single pixel within an area of reduction or by interpolating values between the nearest pixels associated with the centroid of each
new pixel. In biological data, this can overlook considerable variation within the higher resolution data that is not incorporated in the coarser up-scaled data. Careful upscaling of data is extremely important when generating environmental files for SDMs. One new tool aims to provide more flexibility and precision when upscaling rasters by using input spatial statistics, such as mean, majority, median, and minority to name a few.

## DISCUSSION

As previously mentioned, a large proportion of SDMtoolbox v1 users cite using the program for assisting species distribution modeling. One of the reasons for this broad use appears to be the software's balance of user control with the simplicity of the graphical user interface (GUI) and automated model parameterization. SDMtoolbox facilities modeling from start to finish, by aiding compilation of occurrence and environmental data, and spatially vetting occurrence data to reduce spatial biases in occurrence records. Lastly, it rigorously parametrizes each MaxEnt model by performing spatial jacking in conjunction with tuning experiments (adjusting regulation multipliers and feature classes, discussed in itemized points below). This parametrization approach can allow the selection of model settings that have high discriminatory ability and a model that minimizes overfitting to noise as well as to the spatial biases in occurrence data and corresponding environmental biases (*Radosavljevic & Anderson, 2014*). Low overfitting and high discriminatory ability are the two main desired qualities of a 'good' species distribution model (*Lobo, Jiménez-Valverde & Real, 2008*; *Peterson et al., 2011*; *Warren & Seifert, 2011*). Discriminatory ability characterizes the ability of the model to distinguish suitable from unsuitable areas and is typically measured with the area under the curve of the receiver operating characteristic plot (AUC/ROC) (*Peterson et al., 2011*) output from MaxEnt (and other programs). Overfitting is the tendency of a model to fit the random error (or any bias in the sample) rather than the true relationship between the calibration records and predictor variables. Often, over-fit models predict the calibration data very well, but perform poorly on other data sets. Overfitting is typically assessed with the false negative rate, also called omission error-rate (OER henceforth). With an appropriately selected threshold converting a continuous prediction into a binary one, OERs indicate the proportion of presences incorrectly classified as falling into unsuitable areas (typically resulting from a prediction that is too tightly fit to the conditions at calibration localities; *Anderson, 2003*). The best model output from SDMtoolbox is the model with the lowest OER, and of those models, if multiple, the model with the highest AUC. Lastly, if a model has an identically low OER and high AUC values, the feature class complexity is accounted for selecting the model with the lowest complexity. If there is a single model with the lowest OR, AUC values should be assessed in the MaxEnt outputs post-hoc and used as an independent metric of performance resulting from model tuning.

The primary ways SDMtoolbox minimizes model overfitting and properly parameterizes each species distribution model it creates (vs. using only MaxEnt without SDMtoolbox) are discussed in the following paragraphs. The first example concerns the curation of occurrence records. To perform well, most SDM methods require input-occurrence

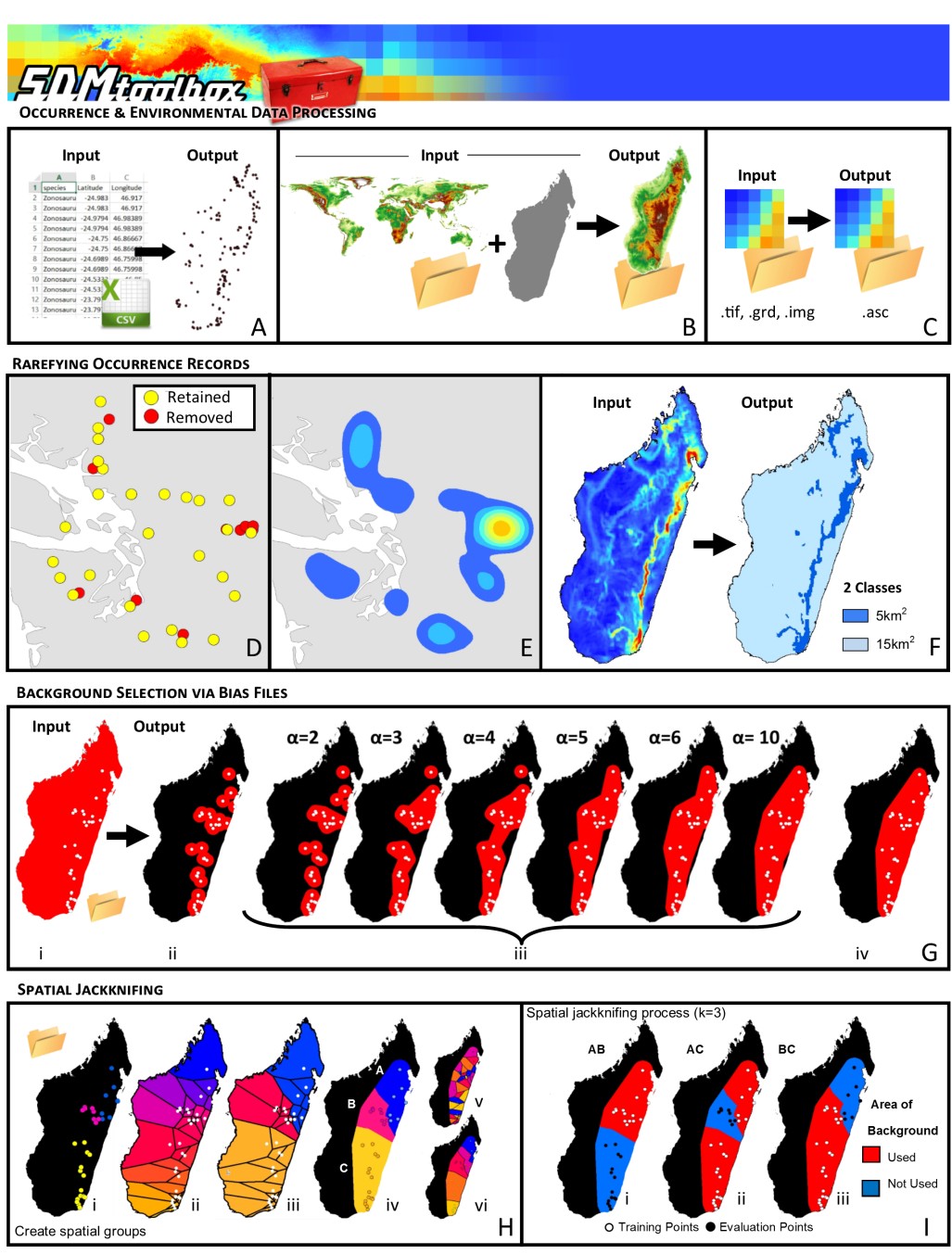

**Figure 1** **Visual Overview of Using SDMtoolbox to model in MaxEnt.** SDMtoolbox box has tools to facilitate input and processing occurrence and environmental data: (A) Convert CSV or XLS files to shapefiles. (B) Clipping environmental layers to spatial extent. (C) & (D) Conversion from raster formats to ASCII for MaxEnt. SDMtoolbox also reduces spatial biases in occurrence record by spatially rarefying the points (a.k.a. spatial filtering) to reduce clusters of points. (E) Areas of high spatially autocorrelated occurrence records that were removed during spatial rarefying. Blue to orange colored polygons depict low to high levels of spatial autocorrelation existing in occurrence records prior to spatial rarefying. (continued on next page...)

**PeerJ** ___________________________________________________

**Figure 1 (…continued)**
(F) The spatial filtering process can be done with a single spatial filter or up to five spatial filters to account for topographic and climatic heterogeneity. For example, in areas of high climate heterogeneity points could be filtered at a smaller area and in areas of low climate heterogeneity at larger spatial scales (i.e., 5 km$^2$ for areas of high and 15 km$^2$ for areas of low heterogeneity). (G) One way SDMtoolbox minimizes model overfitting of each species distribution model it creates is by carefully controlling the background selection using bias files. SDMtoolbox provides several methods for being more selective in the choice of background points in MaxEnt: (i) distance from observation points, (ii) buffered local adaptive convex-hull of observations (a flexible way to create cluster of smaller convex polygons), and (iii) buffered minimum-convex polygon of observation points. (H) Spatial jackknifing tests and evaluating performance of spatially segregated localities. SDMtoobox splits the landscape into 3–5 regions based on spatial clustering of occurrence points (Hi) and classification of clusters into Voronoi polygons (Hii–iii). Users can have between 3–5 spatial random (Hiv) or segregated groups (Hv). Models are calibrated using permutations of training occurrence data from $n-1$ spatial groups, and then are evaluated with the withheld spatial group (I). Here $k = 3$ and each group was label as A,B,C. Models were trained with points from areas two areas, then evaluated with points from the area not included in training. This process continues until models are evaluated with point from each spatial area (e.g., A, B or C) and trained with points from all other areas (e.g., AB, AC or BC, respectively).

data to be spatially independent. However, researchers often introduce environmental biases into their SDMs from spatially autocorrelated occurrence points. It is important to eliminate spatial clusters of localities for model calibration and evaluation. When spatial clusters of localities exist, often models are over-fit towards environmental biases (reducing the model's ability to predict spatially independent data) and model performance values are inflated (Fig. 1D; *Veloz, 2009*; *Hijmans, 2012*; *Boria et al., 2014*). In SDMtoolbox, this can be done in several ways, using the *Spatially Rarefy Occurrence Data for SDMs (reduce spatial autocorrelation)* tool set.

The second way SDMtoolbox minimizes model overfitting of each species distribution model it creates is by carefully controlling the background selection by using bias files. Bias files control where background points are selected and thereby avoid habitats greatly outside of a species' known occurrence. Background points are meant to be compared with presence data to help identify the environmental conditions under which a species can potentially occur. Typically, background points are selected within a large rectilinear area. Within such areas, environmentally suitable but uncolonized or biogeographically isolated habitat often exists. The selection of background points within these habitats increases commission errors (false positives). As a result, the 'best' performing model tends to be over-fitted because the selection criterion favors a model that fails to predict the species in the un-colonized climatically suitable habitat (*Anderson & Raza, 2010*; *Barbet-Massin et al., 2012*). The likelihood that suitable unoccupied habitats are included in background sampling increases with distance from the realized range of the species. Thus, a larger study of spatial extent can lead to the selection of a higher proportion of less informative background points (*Barbet-Massin et al., 2012*). Such issues are ameliorated by being more selective in the choice of background points in MaxEnt (*Barve et al., 2011*; *Merow, Smith & Silander, 2013*). In SDMtoolbox, this can be done several ways, using the *Gaussian Kernel Density of Sampling Localities*, *Sample by Buffered Local Adaptive Convex-Hull*, *Sample by Buffered MCP*, and *Sample by Distance from Obs. Pts.* tools (Fig. 1G).
The MaxEnt program attempts to limit model complexity during parameterization by controlling model regularization. This regularization imposes a penalty for each term included in the model and for higher weights given to a term (*Phillips, Anderson & Schapire, 2006*; *Anderson & Gonzalez, 2011*). The current release of MaxEnt implements a regularization multiplier, which is a user-specified coefficient that is applied to the value of the respective beta parameters of each feature class incorporated into the model (*Phillips, Anderson & Schapire, 2006*). This regularization multiplier alters the overall level of regularization rather than changing each beta parameter individually. The default setting in MaxEnt is a value of 1. Researchers have reported that regularization multipliers as high as 2.0 to 4.0 were necessary to reduce overfitting resulting from lower regularization multiplier values (*Radosavljevic & Anderson, 2014*). Qualitative assessments of the geographical predictions reiterate this conclusion (*Radosavljevic & Anderson, 2014*). To evaluate and compare multiple regularization parameters, in absence of SDMtoolbox, MaxEnt requires users to run multiple instances of the program and then manually compare model performance statistics (i.e., OR and AUC). SDMtoolbox v2.0 allows you to input a range of regularization multipliers and automatically selects the value resulting in the best model (via the process clarified in the first paragraph of the 'Discussion').

Another key parameter in a MaxEnt model is the feature class, which determines the kinds of constraints allowed in a model. A feature is a function of user-input environmental variables and can be any single one or various combination of six feature classes implemented: linear (L), quadratic (Q), product (P), threshold (T), hinge (H) or category indicator (C) (*Phillips, Anderson & Schapire, 2006*; *Phillips & Dudík, 2008*). The MaxEnt model features impose varying constraints on the relationship between the occurrences and user input environmental variables and result in models of varying complexities. It is important to remember that even if multiple feature classes are allowed for model-building, not all classes will necessarily be incorporated in the final model. The default MaxEnt setting for feature class, called "auto features," applies the class or classes estimated to be appropriate for the particular sample size of occurrence records based on extensive tuning experiments (*Phillips & Dudík, 2008*). The use of complex feature settings allows MaxEnt to produce a model that is more sensitive to details of a species' environmental tolerances. However, complex feature classes can also lead to over-fit models. *Phillips & Dudík (2008)* selected the following feature classes for continuous variables as default for the corresponding occurrence record sample sizes: all feature classes for at least 80 occurrence records; L, Q and H for sample sizes 15 to 79; L and Q for 10 to 14 records; only L for below 10 records (*Phillips & Dudík, 2008*). These settings were subsequently implemented in the "auto features" of MaxEnt. However, as recommended by *Shcheglovitova & Anderson (2013)* to reduce model overfitting, SDMtoolbox does not limit feature class use by the number of occurrence records and allows users to compare models created from five different combinations of feature classes: 1. L; 2. L & Q; 3. H; 4. L, Q & H; 5. L, Q, H, P & T. For the fifth group of feature class combinations, in SDMtoolbox v2 the threshold feature class (T) can be enabled or disabled in accordance with the current MaxEnt 3.4.1 recommendations (where T is disabled) or previous versions (where T is enabled). The C class is reserved for categorical variables and independently

applied to associated layers in all five feature class combinations. In addition to the methods implemented in SDMtoolbox, we urge users to view of responses curves in their final model and evaluate if they are biologically logical.

Spatial jackknifing (or geographically structured k-fold cross-validation) tests and evaluates performance of spatially segregated localities. The last step of model parameterization SDMtoolbox implements spatial jackknifing. To do this, SDMtoobox splits the landscape into 3–5 regions based on Voronoi polygons (a polygon whose interior is closest to an individual occurrence record) and spatial clustering of occurrence points (Fig. 1G). Models are calibrated with all permutations of the groups using occurrence points and background data from $n − 1$ spatial groups and then evaluated with the withheld group (Fig. 1H). Spatial jackknifing has demonstrated clear advantages over random sampling of test/training occurrence data (as is common practice and the default setting in MaxEnt). In experiments on the effects of these two treatments, randomly partitioned occurrence datasets produced inflated estimates of performance and led to over-fit models (*Radosavljevic & Anderson, 2014*). Under the spatial jackknifing approach, increasing the regularization multiplier did not sufficiently counteract the effects of the strong spatial bias in the localities used for model calibration (artificially inserted into their experimental approach; *Radosavljevic & Anderson, 2014*). In contrast, the spatial jackknifing approach was shown to sidestep problems of the artificial spatial bias (and any corresponding environmental biases) and allow for detection of overfitting to environmental biases that differed among the spatial partitions (*Phillips, Dudík & Schapire, 2017*). Spatial jackknifing, evaluation of multiple feature classes and multiple regularization parameters are implemented within the *Run MaxEnt: Spatial Jackknifing* tool.

## CONCLUSIONS

The scripts in SDMtoolbox 2.0 streamline many geospatial analyses and simplify GIS processes associated with analyzing biological datasets. SDMtoolbox 2.0 can dramatically reduce the repetitive and time-consuming analyses commonly associated with species distribution modeling: data pre-processing, modeling parametrization, model evaluation, and post-SDM analyses. This release of SDMtoolbox allows researchers to use the most current *ESRI, 2017* software and reduces the amount of time that would be spent developing common solutions. The latest version of SDMtoolbox, a user guide, and example data are freely available at http://www.sdmtoolbox.org. For questions or suggestions regarding SDMtoolbox 2.0, go to our Google Group (https://groups.google.com/forum/#!forum/sdmtoolbox) or email sdmtoolbox.help@gmail.com.

### Funding
The authors received no funding for this work.

## Competing Interests

The authors declare there are no competing interests.

## Author Contributions

- Jason L. Brown conceived and designed the experiments, analyzed the data, edited and evaluated Python code, wrote the paper, prepared figures and/or tables, reviewed drafts of the paper.
- Joseph R. Bennett and Connor M. French analyzed the data, edited and evaluated Python code, reviewed drafts of the paper.

## Data Availability

Python code is contained within the ArcGIS toolbox in the software zip file. Scripts are open source from within ArcGIS. Right click any script in SDMtoolbox and select 'edit'. If prompted for a password, enter 'dendrobates'. This allows code to be accessible by all, but limits accidental edits to code.

## Supplemental Information

Supplemental information for this article can be found online at http://dx.doi.org/10.7717/peerj.4095#supplemental-information.

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
