# Peer review of "SDMtoolbox 2.0: the next generation Python-based GIS toolkit for landscape genetic, biogeographic and species distribution model analyses"

_PeerJ, doi:10.7717/peerj.4095_

## Round 0.1 · original submission · Major Revisions

Dear Authors,

I have now received two reviews, one minor and one major revisions. I am recommending major revision.
Reviewer number 1 found bugs in your scripts and these will need to be correct (the scripts need to run without throwing errors). Reviewer number two had suggestions on orthography of the MS, and reviewer one the inclusion of page numbers in the User Guide.
I would appreciate if you addressed the issue of errors and thoroughly test your scrips (I know this is Python, but different operating systems, functions are sensitive to differences in end of line characters among operating systems, etc.) and implemented the reviewers’ comments and suggestions before returning a revised MS.
I look forward to your revision.
Sincerely,

Tomas Hrbek

Reviewer 1 ·

Basic reporting

This manuscript provides detailed documentation concerning the release of a new version of SDMtoolbox: 2.0. As the use of geospatial analyses is continually increasing in many ecological fields, the development and introduction of new tools such as this that can be used for robust study design is highly important. Generally, this manuscript and the associated “Getting Started” and “ User Guide” are clear and well written. However, I have provided detailed edits in the attached document that can improve organization and clarity of the manuscript, getting started, and user guide documents. The User Guide needs page #s to match the Table of Contents. Additionally, some of the language is unprofessional and needs to be changed as I noted in the detailed edits. Overall, the User Guide was straight-forward, however I received some toolbox errors when following the User Guide as closely as possible, and I have included the error logs for reference. I think that if a better description of the files that are being used for each step and their relative location are added, these errors may be avoided. Finally, there is a lot of detail in the User Guide concerning Maxent, however a description of the new Clog Log function is absent; this should be added.

Experimental design

N/A

Validity of the findings

N/A

Annotated reviews are not available for download in order to protect the identity of reviewers who chose to remain anonymous.

Reviewer 2 ·

Basic reporting

Please follow correct citation style (sometimes publication year is within brackets and other times it is not)

Experimental design

Ok

Validity of the findings

Ok

Additional comments

Reviewer 1:
SDMtoolbox 2.0: The next generation python-based GIS toolkit for landscape genetic, biogeographic and species distribution model analyses.
I congratulate the authors for creating and updating SDMtoolbox 2.0. The toolbox aids the automation of rigorous spatial analyses; thus, it is highly desirable among geospatial modelers and conservation professionals. My recommendation is to accept with minor revision. I encourage the authors to make suggested changes and resubmit for publication. Here are my comments.
Detailed Comments
Line 36, a total of 79 scripts
Line 40, future climates,
Line 55, a series of
Line 56, complicated spatial analyses in ArcMap (ESRI) and python
Line 55, line 108, and others, please write abbreviations in full at first use.
Line 74, Rewrite the last sentence.
Line 89, e.g.,
Line 101, importance of Bias files in Maxent
Line 117, delete brackets, ..tools batch project or define projections of input rasters….
Line 118, delete ‘aims to’, another set of tools facilitate batch exportation of images or aid in RGB band visualization of multiband rasters
Line 156, rewrite. Sentence starting As following is not grammatically correct.
Line 163, make sure all figures are mentioned in the text.
Line 164,insert ‘in’, it can be done in several ways
Line 191, “lower levels”, did you mean resulting from lower regularization multiplier values
Line 196, delete “(i.e. Bio1)”
Line 205, that is more sensitive details of ? sentence appears incomplete
Line 207, “which subsequently is implement in the”, sentence appears incorrect
Line 217, tests and evaluates?
Line 241, if the journal allows this should be written as footnote or in the acknowledgment section.
Please follow correct citation style (sometimes publication year is within brackets and other times it is not)
I installed the tool using the software installation guide provided in the supplemental material. The user guide needs some editing as well. Example, first sentence needs the preposition “to”. It should read like “designed to automate”. The second paragraph needs a comma immediately after SDMtoolbox 2.0.
Table of contents lists page 215 while the document is only 95 pages.
General comments:
Line 213, the latest maxent default combination is: L, Q, P, H which appears to be more appropriate than the previous maxent versions that included all of them (i.e, L,Q,P,H and T). Are the users able to use this new maxent default combination in SDMtoolbox 2.0?
The methods section is very brief, can the authors expand the methods section a little bit.
ESRI is an institution that frequently changes its software versions. Given this fact, what is the authors long-term plan to keep SDMtoolbox running on new ArcGIS software versions? Can SDMtoolbox become a standalone tool?

---

## Round 0.2 · Minor Revisions

Dear Authors,

Thank you for your thorough and thoughtful responses to the reviewer’s comments.
I am returning the MS to you so that you can implement some of the small edits suggested by reviewer 2. With respect to the use of “Thou shall” the choice is yours, although I too would prefer to see alternate wording.
I look forward to your revision.
Sincerely,

Tomas Hrbek

Reviewer 1 ·

Basic reporting

This manuscript provides detailed documentation concerning the release of a new version of SDMtoolbox: 2.0, and will be an excellent contribution to PeerJ. Following the edits and suggestions of the reviewers, the authors have greatly improved the manuscript and the associated “Getting Started” and “ User Guide” for the toolbox. They also made changes to the script to correct previous errors. I fundamentally disagree on the continued use of "The 10 commandments of SDMtoolbox 2.0", especially by beginning each line with "Thou shall". This is, in my opinion, the only unprofessional part of the document. I have no further edits or suggestions.

Experimental design

No comment

Validity of the findings

No comment

Reviewer 2 ·

Basic reporting

Ok

Experimental design

Ok

Validity of the findings

Ok

Additional comments

I have minor comments.
The discussion section reads well but other sections need additional work.
Line 111, sentence not clear, rewrite.
Line 122, sentence not clear, particularly use of the verb visualizing
Line 130, incorrect use of punctuation marks. Delete the colon and all semi-colons, and replace the last phrase with "to name a few". Or rewrite.
Line 155, a coma is need after OR. Or rewrite
Line 203, delete (i.e mean annual temperature)

79 scripts in the main document vs 78 in the user guide. Which is correct?
User guide table of content still refers to page 215

---

## Round 0.3 · accepted · Accept

Dear Jason,

I am happy to accept you manuscript for publication in PeerJ. Congratulations!

Cheers,

Marcial.